# Influence of Leaf Area Index Inversion and the Light Transmittance Mechanism in the Apple Tree Canopy

Linghui Zhou, Yaxiong Wang, Chongchong Chen, Siyuan Tong and Feng Kang *

School of Technology, Beijing Forestry University, No. 35 Qinghua East Road, Beijing 100083, China; zhoulinghui@bjfu.edu.cn (L.Z.); yaxiongwang87@bjfu.edu.cn (Y.W.); chenchongchong@bjfu.edu.cn (C.C.); tongsiyuan@bjfu.edu.cn (S.T.)
* Correspondence: kangfeng98@bjfu.edu.cn; Tel.: +86-010-62336137-709

**Abstract:** Light plays a crucial role in the growth of fruit trees, influencing not only nutrient absorption but also fruit appearance. Therefore, understanding fruit tree canopy light transmittance is essential for agricultural and forestry practices. However, traditional measurement methods, such as using a canopy analyzer, are time-consuming, labor-intensive, and susceptible to external influences, lacking convenience and automation. To address this issue, we propose a novel method based on point clouds to estimate light transmittance, with the Leaf Area Index (LAI) serving as the central link. Focusing on apple trees, we utilized handheld LiDAR for three-dimensional scanning of the canopy, acquiring point cloud data. Determining the optimal voxel size at 0.015 m via standardized point cloud mean spacing, we applied the Voxel-based Canopy Profile method (VCP) to estimate LAI. Subsequently, we established a function model between LAI and canopy light transmittance using a deep neural network (DNN), achieving an overall correlation coefficient $R^2$ of 0.94. This model was then employed to estimate canopy light transmittance in dwarfed and densely planted apple trees. This approach not only provides an evaluation standard for pruning effects in apple trees but also represents a critical step towards visualizing and intelligentizing light transmittance.

**Keywords:** 3D point cloud; Leaf Area Index; light transmittance; voxel

## 1. Introduction

China is a major producer of apples, accounting for approximately 50% of the world's supply. The canopy has a significant impact on fruit parameters and quality. Canopy lighting is the most important factor. Light transmittance is a measure of the amount of light that penetrates the leaves, which reflects the intensity of photosynthetic radiation absorbed by the plant canopy and the distribution of nutrients in the leaves [1]. Light transmittance has a direct impact on plant growth, development, and photosynthesis in the canopy. It is also crucial for understanding the structure and function of plant leaves and the ecological and physiological processes in plant ecosystems [2]. Measurement of light transmission is primarily instrumental, involving considerable time, labor, and subjectivity.

In the study of canopy light transmittance, many findings indicate that light transmittance is related to the Leaf Area Index (LAI). This applies to related research on various agricultural and forestry crops such as corn [3], pine, beech [4], fir [5], and birch [6]. However, at present, a specific functional model between light transmittance and the Leaf Area Index (LAI) has not been established, and understanding the mechanism of how LAI impacts light transmittance remains crucial for enhancing the intensity of photosynthetic radiation received by the canopy. The advancement of photosynthetic radiation measurement methods and 3D reconstruction technology has provided technical support for establishing the functional relationship between light transmittance and LAI. In general, orchard point cloud information can be obtained not only through LiDAR but also through imagery. The advantage of image methods lies in the simplicity of the shooting process. However, there

are several limitations to consider, including perspective constraints, shadow interference, and reflection issues. Furthermore, extracting in-depth information during data processing adds to the workload. In contrast, LiDAR is unaffected by lighting conditions during point cloud data collection, making it a more direct and accurate method of data collection.

The spatiotemporal variability of light is significant, making it difficult to measure the light environment of the canopy. To date, researchers have proposed many methods to measure canopy light transmittance, mainly including methods based on measuring photosynthetic radiation with instruments, methods based on ray tracing, and methods that combine LiDAR scanning with the Beer–Lambert law.

Many researchers have used instrument-based measurements of photosynthetic radiation. For instance, Hale et al. [7] utilized DHP to gauge canopy light transmittance in Scots pine forests and modeled it with their stand parameters. Hossain et al. [8] estimated canopy light transmittance in cedar hemlock using DHP and LAI-2200, respectively, while taking into account the effects of stand characteristics and weather conditions. Similarly, PAEKER et al. [9] measured light transmittance in forests of different ages and canopy types using quantum sensors. The method based on measuring photosynthetic radiation with instruments is relatively quick and convenient, but it requires a high-quality surrounding environment. For instance, when using digital hemispherical photography (DHP), external environmental factors (such as the presence of shadows, instrument parameter selection, and dust in the air) can introduce noise into the images [10]. When using LAI-2200 under poor lighting conditions, there may be issues with light obstruction, and it is necessary to use a view cap to eliminate the adverse factors of unequal sky conditions (clouds, open spaces, or branches), which leads to increased operational complexity and the introduction of subjective selection variability [11]. Moreover, systematic errors produced by instruments like light meters and quantum sensors occur due to the deviations in specific spectral sensitivity and efficiency of embedded multifunction sensors, and they also depend on various types of radiation [12]. Additionally, experimental work requires a large amount of sampling data and numerous readings, which is both time-consuming and labor-intensive.

The ray-tracing approach models the transmission, reflection, and incidence of solar beams within the canopy. Bittner et al. [13] conducted light simulation of beech seedling canopies by combining a three-dimensional canopy structure with a fast ray-tracing algorithm. However, this method necessitates sampling a substantial amount of light to capture the impact of each individual in the canopy. Studies have indicated that the computational complexity of this approach increases exponentially with the level of geometric detail in the reconstructed canopy model and the number of simulated emitted solar rays in the target scene [14]. Therefore, ray-tracing-based approaches can be computationally intensive when addressing complex lighting scenarios within the tree canopy. It is essential to recognize that this method may not always offer the most efficient solution [15].

The method based on LiDAR scanning combined with the Beer–Lambert law utilizes laser beams instead of sunlight to estimate light transmittance. This is achieved by simulating the penetration ratio of the laser through the canopy. Musselman et al. [16] developed a light transmittance model based on Beer's law using LiDAR and canopy indicators. However, this type of model fails to account for detailed factors, leading to occlusion issues and difficulty in describing canopy structure intricately. Consequently, it lacks precision in depicting structural variables of vertical multi-layer forests and falls short in adequately addressing incident and transmitted solar radiation at the scale of individual forest stands. Moreover, the incidence angle of these beams remains static and cannot be dynamically adjusted over time [17].

The Leaf Area Index (LAI) is a crucial metric representing the density of plant canopies, defined as the ratio of total leaf surface area to the ground area it covers. This parameter plays a pivotal role in characterizing vegetation structure and is closely associated with fundamental processes like photosynthesis, transpiration, and respiration. LAI has emerged as a fundamental biophysical variable in disciplines such as agriculture, forestry, ecology, and meteorology. Its significance lies in its relationship with crop growth, canopy light

absorption, and its frequent application in the development of plant growth models, energy balance models, and canopy reflectance models.

The acquisition of Leaf Area Index (LAI) involves both experimental measurement and numerical calculation methods. Experimental methods for measuring LAI values can be categorized as direct and indirect measurements. Direct measurement techniques include collecting fallen leaves [18] or employing destructive sampling methods, where LAI is determined manually. However, these methods are highly destructive, irreversible to the plants, labor-intensive, and often yield relatively low accuracy. In contrast, indirect measurement methods rely on optical instruments. Currently, commonly used LAI measuring instruments include digital hemispherical photography, the Plant Canopy Analyzer LAI-2200, and the AccuPAR light interception device. These instruments offer high measurement accuracy and do not cause damage to the canopy.

Numerical calculation primarily involves obtaining point cloud information through LiDAR and deriving the canopy Leaf Area Index from the point cloud data. Currently, available LiDAR-based methods for estimating the Leaf Area Index include the regression model method, gap fraction method, and voxel method. In the regression model method, LiDAR is utilized to capture vegetation parameters such as the tree height and diameter at breast height, which are then used to derive regression equations for the Leaf Area Index [19]. The gap fraction method characterizes the probability that laser beams pass through the crop canopy without interception, effectively distinguishing between non-intercepted and intercepted laser pulses. This method has been applied to calculate the Leaf Area Index (LAI) for various tree species, including birch, eucalyptus [20], and larch [21].

The voxel method, utilized for calculating the leaf area density, primarily relies on the contact frequency of lasers within the point cloud. By integrating the leaf area density, the Leaf Area Index (LAI) can be derived. Hosoi and Omasa introduced the profile analysis method, which incorporates corrections for leaf inclination and non-photosynthetic tissue, effectively reducing estimation errors. Their research suggested that for unknown actual leaf inclination angles, a laser beam incidence zenith angle close to $57.5°$ yields better correction effects. This methodology has been applied across various plant species, including camellia [22], beech [23], wheat [24], and rapeseed [25], to explore the relationship between canopy indicators and yield. Van et al. [26] employed this method to calculate LAI values for tree species like beech, London plane, and pre-grass. Based on their studies, they proposed a light interception model to estimate average light distribution throughout the season. Li et al. [27] employed a Gaussian mixture model to segment point cloud leaves of magnolia trees and calculated the vertical leaf area density (LAD) profile and LAI values of the canopy.

Studies have focused on the relationship between the Leaf Area Index (LAI) and light transmittance (LT). It is important to note that while LAI has been extensively studied, other factors also affect light transmittance. For instance, factors such as the size, shape, and inclination angle [28] of leaves can significantly influence canopy light transmittance. While the size and shape of leaves tend to be consistent within the same tree species, the leaf inclination angle remains a key factor affecting light transmittance. Moreover, the physiological condition of vegetation, including factors like the water status and chlorophyll content, along with atmospheric conditions and seasonal variations, can also impact canopy light transmittance [29]. However, among these influencing parameters, the Leaf Area Index (LAI), which reflects canopy density, stands out as the most significant determinant of light transmittance. Despite its importance, the specific mechanism underlying the influence of LAI on light transmittance remains unclear, warranting further exploration and development of a theoretical model to elucidate their relationship.

In summary, this study aimed to utilize LiDAR point cloud data and a voxel-based canopy contour (VCP) modeling approach to invert the canopy Leaf Area Index. The optimal voxel size for calculating the canopy Leaf Area Index was determined to be the standard Euclidean distance. Additionally, canopy light transmittance was measured using a canopy analyzer, and the Leaf Area Index was modeled as a function of this measurement.

While this method can be time-consuming and labor-intensive, it provides an objective evaluation of canopy light transmittance, circumventing subjective factors. It allows for scanning a larger area of an orchard without being constrained by external conditions, thereby reducing the time and labor needed for accurate assessments. This enhances efficiency for precise orchard research and simplifies measurements of light transmittance.

## 2. Materials and Methods

### 2.1. Selection of Test Site

The study site is located in Zangjiazhuang Town, Fushan District, Yantai City, Shandong Province (Latitude: 37°46′, Longitude: 120°99′). This area is one of the most representative apple production bases in China, covering an area of 2000 square meters. The region has a temperate continental monsoon climate, with an average altitude of approximately 65 m, an average annual temperature of 12.6 °C, an average annual precipitation of 529 mm, and an average annual sunshine duration of 2489 h.

The orchards in the region predominantly feature dwarf and dense planting patterns of Fuji apple varieties. The trees are spaced 2.0 m apart with a row spacing of 4.0 m. The breast height diameter of the trees is approximately 0.15 m, with a canopy width of 3.5 m by 4.0 m and a tree height of approximately 3.0 m. For this study, 35 twelve-year-old apple trees were selected as research subjects. Figure 1 shows the specific location and growth conditions of the orchard.

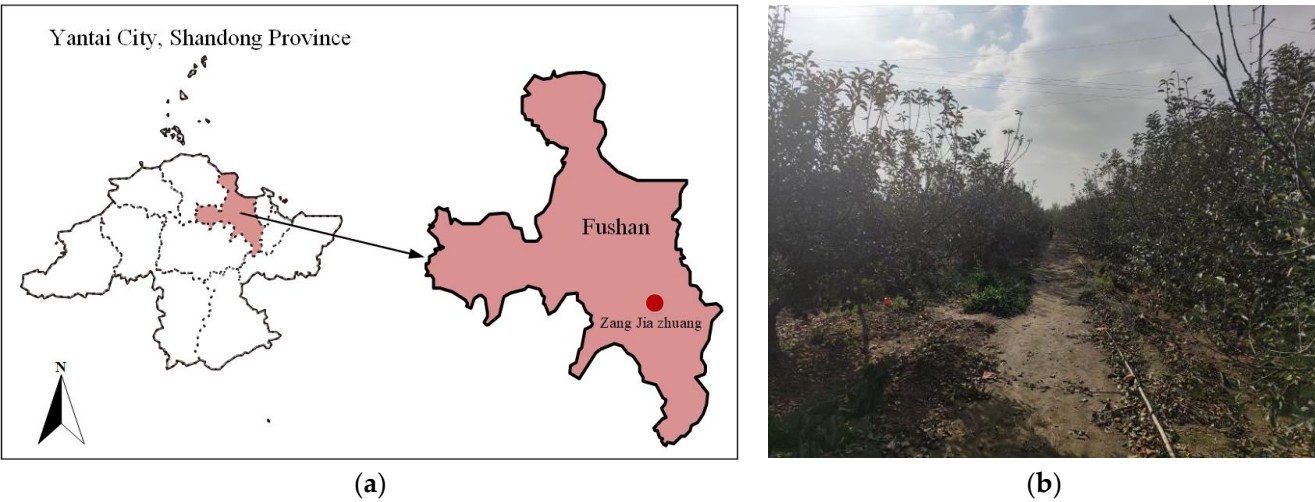

(**a**)         (**b**)

**Figure 1.** Experimental scenario diagram. (**a**) Specific location of the orchard. (**b**) Partial orchard situation.

### 2.2. Collection Equipment

2.2.1. Point Cloud Data Acquisition Equipment

In this study, a handheld LiDAR system (3D-BOX) was used to obtain 3D point cloud data on trees. The device, as depicted in Figure 2, comprises a 3D laser scanner, an IMU inertial measurement unit, and a microcomputer [30]. A VLP-16 LiDAR 3D laser scanner, manufactured by Velodyne Lidar Co., San Jose, CA, USA, and an Mti-30-2A8G4 IMU, manufactured by Xsens Co., Enschede, Holland, were used in this study. The specific parameters of the VLP-16 LiDAR are listed in Table 1.

The equipment connections are represented by dashed lines, while solid lines indicate connections to the power supply system.

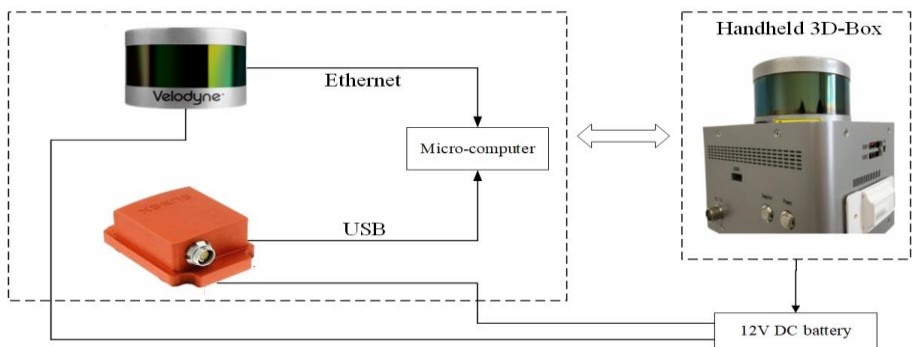

**Figure 2.** Handheld LiDAR system.

**Table 1.** LiDAR parameter table.

| Feature | Parameter | Feature | Parameter |
|---|---|---|---|
| Scanning range | 0.1~100 m | measurement accuracy | ±3 cm |
| Horizontal field of view angle | 360° | Vertical field of view angle | 30° |
| Horizontal angle resolution | 0.1°~0.4° | Vertical angle resolution | 2° |
| Laser level | 1905 nm | Scanning frequency | 5~20 Hz |
| Number of laser lines | 0.1~100 m | Working voltage | 9~32 V |

### 2.2.2. Canopy Light Transmittance Acquisition Equipment

In this study, a canopy analyzer (LD-G20H, Shandong Lainde Intelligent Technology Co., Ltd., Liaocheng, China) was used for the determination of the light transmittance of the canopy. The device comprises a fish-eye image capture probe and a measuring rod with 25 built-in standard rod count sensors, as shown in Figure 3. The fish-eye image capture probe includes a fisheye lens and a CCD image sensor, with specific parameters detailed in Table 2. The power supply system consists of an 8.4 V lithium battery. Data analysis was conducted using image analysis software and image acquisition software.

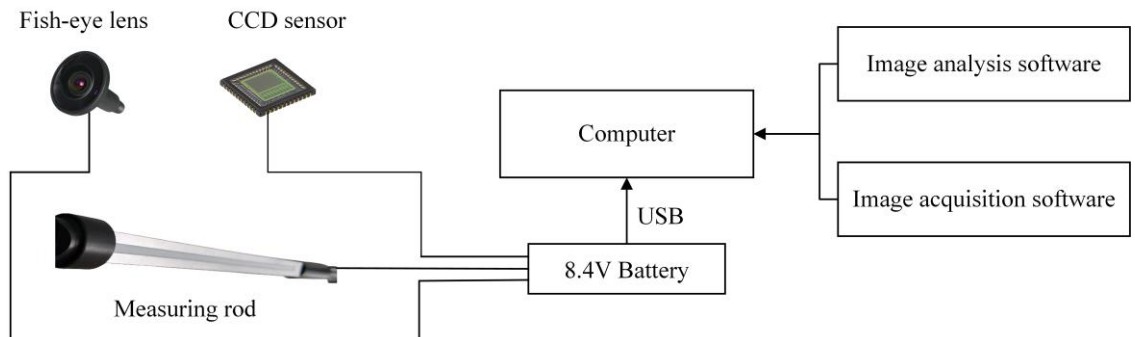

**Figure 3.** Canopy analyzer system.

**Table 2.** Canopy analyzer parameter table.

| Feature | Parameter | Feature | Parameter |
|---|---|---|---|
| Lens angle | 150° | Resolving power | 768 × 494 pix |
| PAR sensing range | 400 nm~700 nm | Measuring range | 0~2000 μmol/m²·S |
| Working voltage | 8.4 V | Working temperature | 0~55 °C |

### 2.3. *Experimental Scheme*

#### 2.3.1. Point Cloud Data Acquisition Experimental Scheme

In this experiment, point cloud data were collected from 35 fruit trees in an orchard. The experimental fruit trees in the orchard were arranged in two rows, and the data were

collected by sequentially scanning from the midpoint placement of the experimental area (black dot) along the direction indicated by the blue and red arrows. The experimental route is shown in Figure 4. Considering the growth of fruit trees, the equipment was placed at a height of about 2.5 m above the ground. Data acquisition was performed at a rate of 0.6 m/s.

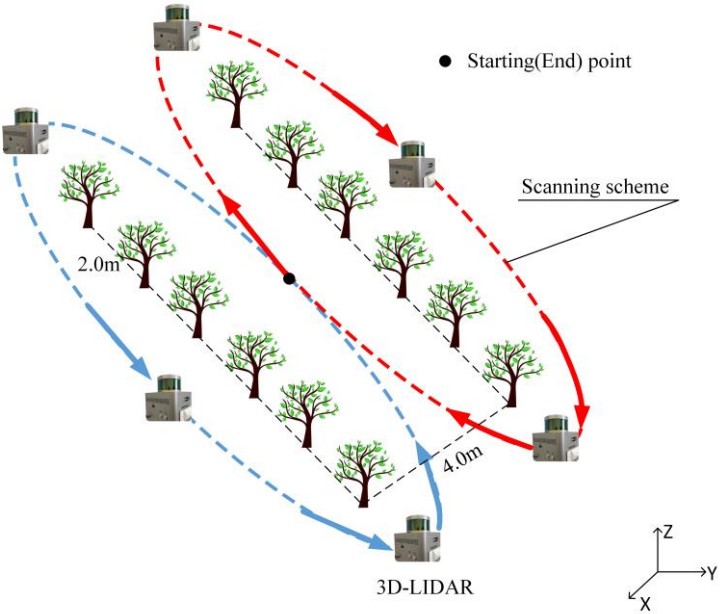

**Figure 4.** Point cloud scanning roadmap.

### 2.3.2. Light Transmittance Data Collection Experimental Scheme

To minimize measurement errors, experiments measuring light transmission should be conducted at the same time every day to ensure uniform light intensity, angle, temperature, and wind conditions. Figure 5 depicts a top view of the collection process, with observation points (red dots) established in four directions at the base of the canopy. The instrument was positioned 0.2 m from the trunk to provide a top-down view of the experimental collection process. It was ensured that the camera lens was positioned vertically and close to the ground while filming. Data collection was conducted five times at each observation point. After completing data collection, the data obtained from four directions were merged to eliminate the influence of tree trunks on light transmittance data.

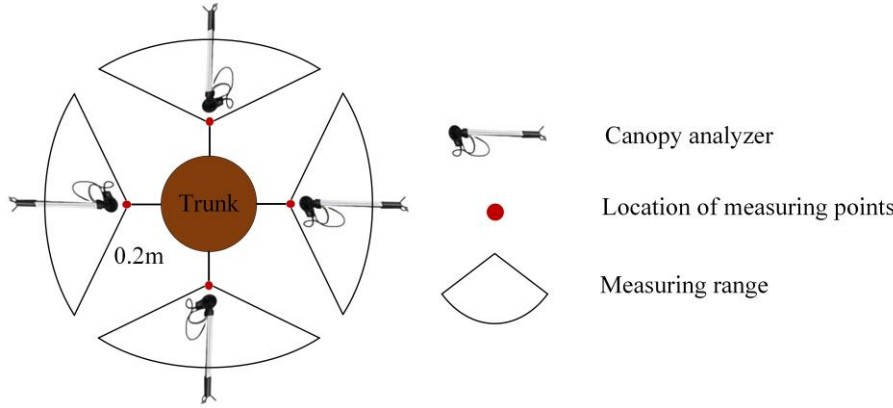

**Figure 5.** Top view schematic diagram of light transmittance acquisition.

*2.4. Data Processing*

2.4.1. Preprocessing of Point Cloud Data

To reduce computational costs and ensure post-study accuracy, the collected point cloud data were pre-processed. This included denoising, filtering, and alignment. The preprocessing process for the canopy point cloud in this paper involved segmenting the overall test area and individual fruit trees using the open-source software CloudCompare (vers. 2.9.1, General Public License software). Due to the uneven ground in the apple orchard, the Cloth Simulation Filter (CSF) [31] was then applied to filter out the ground. Since the main trunk of dwarf apple trees tends to be relatively thick, it is necessary to remove it. This can be achieved using a density-based clustering algorithm to separate the main trunk from the canopy.

2.4.2. Construction of Voxel Model

The voxel algorithm [32] is a method that discretizes continuous point cloud data into a three-dimensional voxel representation. This algorithm maps the point cloud data onto a discrete three-dimensional grid, dividing the space into regular cubic units, where each cubic unit is a voxel.

To establish the VCP (Voxel Canopy Profile) model, it is first necessary to determine the boundary region of the canopy point cloud. The minimum ($X_{min}$, $Y_{min}$, and $Z_{min}$) and maximum ($X_{max}$, $Y_{max}$, and $Z_{max}$) values in the $X$, $Y$, and $Z$ dimensions of the coordinate system are set as the starting and ending points, respectively. The voxel size is set as the step length $v$, and the entire fruit tree canopy is divided into $N_x \times N_y \times N_z$ voxels. The calculation formula is as follows.

$$\begin{cases} N_x = \frac{(X_{\max} - X_{\min})}{v} \\ N_y = \frac{(Y_{\max} - Y_{\min})}{v} \\ N_z = \frac{(Z_{\max} - Z_{\min})}{v} \end{cases} \tag{1}$$

After establishing the VCP model, each small cube is evaluated to determine whether it contains point cloud data. Different characteristic values are then assigned accordingly. If the number of laser points within a voxel is greater than or equal to 1, the voxel characteristic is marked as 1, indicating that the laser beam is intercepted. Otherwise, the voxel characteristic is marked as 0.

2.4.3. Leaf Area Index Inversion Model

The Leaf Area Index (LAI) is calculated using a voxel-based canopy profile method, and the basic formula is as follows [22]:

$$LAD(h, \Delta H) = \alpha(\theta) \cdot \frac{1}{\Delta H} \sum_{\Delta H}^{m_h + \Delta H} N(k) \tag{2}$$

In the formula, $\theta$ represents the average laser incident angle, $\alpha(\theta)$ represents the correction coefficient for the leaf inclination angle, $\Delta H$ represents the thickness of the horizontal layer, $m_h$ and $m_h + \Delta H$ are the voxel coordinates on the vertical axis of the canopy, and $N(k)$ represents the contact frequency of the laser beams within the $k$-th layer.

There is a quantitative relationship between the Leaf Area Index (LAI) and leaf area density (LAD) [33]:

$$LAI = \int_0^H LAD dz \tag{3}$$

Leaf inclination angle correction essentially refers to the correction coefficient for the angle between the leaf inclination and the direction of the laser beam, which is determined by the laser zenith angle of incidence and the leaf inclination angle. The leaf inclination

angle refers to the angle between the normal to the leaf surface and the zenith, ranging from $0°$ to $90°$. The distribution of leaf inclination angles can affect the interception of laser beams by the canopy. The zenith angle of incidence is the angle between the incident laser beam and the direction perpendicular to the ground, obtained through the conversion between Cartesian coordinates and polar coordinates. The conversion formula [34] is as follows:

$$\begin{cases} \gamma = \sqrt{x^2 + y^2 + z^2} \\ \beta = \tan^{-1} \frac{z}{\sqrt{x^2+y^2}} \\ \alpha = \tan^{-1} \frac{x}{y} \end{cases} \tag{4}$$

$x$, $y$, and $z$ are the Cartesian coordinates of the point, and $\gamma$, $\beta$, and $\alpha$ are the polar coordinates of the point, with $\beta$ being the angle between the instrument's scanning direction and the horizontal line.

To calculate the leaf inclination angle, plane fitting based on eigenvalues is employed. The process involves several steps: first, collecting point cloud data to represent the shape of the plane to be fitted; next, calculating the covariance matrix of the data to describe the distribution of the data in each direction; finally, performing eigenvalue decomposition of the covariance matrix to obtain the eigenvalues and corresponding eigenvectors. The eigenvectors represent the main directions of the data distribution, while the eigenvalues represent the degree of dispersion of the data in the direction of the eigenvectors. Among the eigenvalues, only the smallest few and their corresponding eigenvectors are selected to determine the normal vector of the plane. These selected eigenvectors are then used to create the plane equation, typically expressed as a point normal. The fitting effect is evaluated based on the fitted plane parameters, such as calculating the fitting error or verifying that the fitting results meet expectations. Finally, the original point cloud data are utilized to visualize the fitted planes and assess the fitting effect in an intuitive manner.

$\alpha(\theta)$ represents the correction factor affecting the leaf inclination angle when the laser zenith angle is $\theta$, and $G(\theta)$ represents the average projection on the plane perpendicular to the direction of the laser beam for a unit leaf area.

$$\alpha(\theta) = \frac{\cos \theta}{G(\theta)} \tag{5}$$

Assuming that the leaf orientation is symmetrical, the determined $G(\theta)$ is as follows [35]:

$$G(\theta) = \frac{1}{2\pi} \int_0^{2\pi} \int_0^{\frac{\pi}{2}} g(\psi)|\cos(n_B, n_L)|d\psi d\zeta = \int_0^{\frac{\pi}{2}} g(\psi)S(\theta, \psi)d\psi \tag{6}$$

where

$$S(\theta, \psi) = \begin{cases} \cos \theta \cos \psi, \theta \geq \frac{\pi}{2} - \psi \\ \cos \theta \cos \psi \left[1 + \frac{2(\tan x - x)}{\pi}\right], \theta \geq \frac{\pi}{2} - \psi \end{cases} \tag{7}$$

$$x = \cos^{-1} \theta(\cot \theta \cos \psi) \tag{8}$$

In the formula, $\theta$ represents the zenith angle of incidence of the laser beam, $\psi$ represents the leaf inclination angle, and $\varphi$ and $\zeta$ represent the azimuth angles of the laser beam and the normal vector of the leaf surface, respectively. $|\cos(n_B, n_L)|$ denotes the absolute value of the cosine of the angle between the unit vectors of the corresponding laser beam incidence direction and the leaf surface normal vector. $S(\theta, \psi)$ represents the average value relative to the azimuth angle of the leaf surface normal vector. $g(\psi)$ is the distribution function of the leaf inclination angle, which, under the assumption of azimuth symmetry,

is independent of the azimuth angle of the leaf surface normal vector. Based on the actual measured leaf inclination angle [36] is represented as

$$G(\theta) = \sum_{q=1}^{T_q} g(q)S(\theta, \psi) \tag{9}$$

In the formula, $q$ represents the category of different leaf inclination angles, $T_q$ is the total number of leaf inclination angle categories, and $g(q)$ represents the probability distribution of the leaf inclination angle for category q, which is the ratio of the leaf area for category q to the total leaf area.

The contact frequency $N(k)$ refers to the frequency at which the laser beam is intercepted while passing through the fruit tree canopy when collecting data with LiDAR. The formula for contact frequency is as follows:

$$N(k) = \frac{n_l(k)}{n_l(k) + n_p(k)} \tag{10}$$

$n_l(k)$ represents the number of voxels capturing the laser beam at the $k$-th horizontal height layer (with the feature value marked as 1), $n_P(k)$ represents the number of voxels penetrated by the laser beam at the kth horizontal height layer (with the feature value marked as 0), and $n_l(k) + n_P(k)$ refers to the total number of incident laser beams reaching the kth height layer.

Determining the canopy boundary and excluding null elements are crucial steps in calculating the contact frequency using the voxel-based approach, owing to the regularity of the cube. Prior to computing the contact frequency, the outer contour of the crown is identified for each horizontal layer using a 2D convex hull algorithm [37]. The algorithm description is as follows:

(1) To find the starting point, $p_0$, locate the point with the smallest $Y$-axis value. If there are multiple points with the same smallest $Y$-axis value, choose the one with the smallest $X$-axis value as the reference point.

(2) Next, sort the remaining points based on their polar angle from the origin $p_0$. If two points form the same angle with $p_0$, prioritize the one closer to $p_0$. Finally, proceed with a sequential scan of the sorted points starting from $p_0$. If these points are on the convex polygon, then the three consecutively obtained points $p_i - 1$, $p_i$, $p_i + 1$ should satisfy the following property: $p_i + 1$ is on the left side of the vector $<p_i - 1$, $p_i>$. If this property is not satisfied, then $p_i$ must not be a vertex on the convex hull and is deleted.

(3) When $p_i = p_0$, the figure is closed, and the convex polygon is complete. Use the two-dimensional convex hull algorithm to obtain the projection of the canopy point cloud's exterior outline. The laser beams are intercepted by the leaves at the thickness of the horizontal layer, forming a closed convex polygon by connecting the vertices of the convex hull.

Figure 6 illustrates a schematic of the outer contour of tree No. 21, with the boundary depicted by the red line and the black points positioned along the boundary in the point cloud.

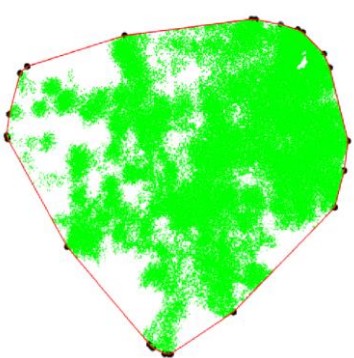

**Figure 6.** Outline diagram of tree No. 21.

2.4.4. Selection of Voxel Size

The size of the voxel can influence the detail of the extracted canopy structure and the accuracy of the contact frequency calculation in the model, making it a key parameter for obtaining canopy structural information. A more optimal voxel model can be obtained by utilizing the average distance between point clouds [27]. There are numerous methods to compute this distance, with the measurement of Euclidean distance being the most prevalent in spatial structure problems. However, the basic Euclidean distance treats variations in dimensions or variables equally, potentially leading to errors in application. Therefore, it is essential to standardize each component.

$$X^* = \frac{X - m}{s} \tag{11}$$

$m$ is the mean of different components and $s$ is the standard deviation of different components, reflecting the degree of dispersion of data in each dimension, $X^*$ is the standardized point cloud data, with a mean of 0 and a variance of 1.

The formula for the standardized Euclidean distance between two points in three-dimensional point cloud data A(x1, y2, z3) and B(x1′, x2′, x3′) is as follows:

$$d^* = \sqrt{\sum_{k=1}^{3} \left( \frac{x_k - x_k'}{s_k} \right)^2} \tag{12}$$

where $d^*$ is the standardized Euclidean distance.

The determination of voxel size in this section is closely tied to the type of plant being studied. As the plant type varies, so does the distribution of the corresponding point cloud, leading to variations in the optimal voxel size.

2.4.5. Leaf Area Index–Light Transmittance Fitting

This paper utilizes a fully connected deep neural network (DNN) to model light transmittance in the canopy. The fully connected DNN model follows a typical multi-layer perceptron (MLP) network structure. In this model, each layer of neurons establishes connections with all neurons in the previous layer, forming a fully connected topology. This architecture aids in extracting deeper features from input data, thereby enhancing the model's representation and generalization capabilities.

The input variables consist of the Leaf Area Index for 35 apple trees, while the output is the light transmittance. In neural networks, the input layer serves as the initial layer of the network, responsible for receiving external input data and passing it to the subsequent layer. Nodes in the input layer correspond to features of the input data. Hidden layers, positioned between the input layer and the output layer, perform nonlinear transformations and extract features from the input data. Each node in the hidden layers receives inputs from the preceding layer, computes them with weights and activation functions, and forwards the results to the subsequent layer. The primary function of the hidden layer is to

learn higher-order features of the input data, enabling the neural network to better adapt to complex data patterns and relationships. The output layer, serving as the final layer of the neural network, produces the network's output. The number of nodes in the output layer typically varies based on the task type, whether it is a classification or regression task. Each output node maps the output of the hidden layer to the final output using weights and activation functions.

The essence of the DNN lies in determining the appropriate number of hidden layers and nodes within each layer. The selection of the number of hidden layers is typically guided by empirical formulas [38,39]:

$$h_l = a + \sqrt{b_i + b_p} \tag{13}$$

*a* is an adjustment variable between 1 and 10, $b_i$ is the number of input variables, and $b_p$ is the number of output variables.

Deep neural networks were employed to model both the Leaf Area Index and canopy transmittance. The final function model is illustrated in Figure 7:

(1)  There are three hidden layers situated between the input layer and the output layer, with 4, 12, and 8 nodes, respectively.
(2)  Logsig, Tansig, and Purelin are utilized as transfer functions from input to output for the three hidden layers.

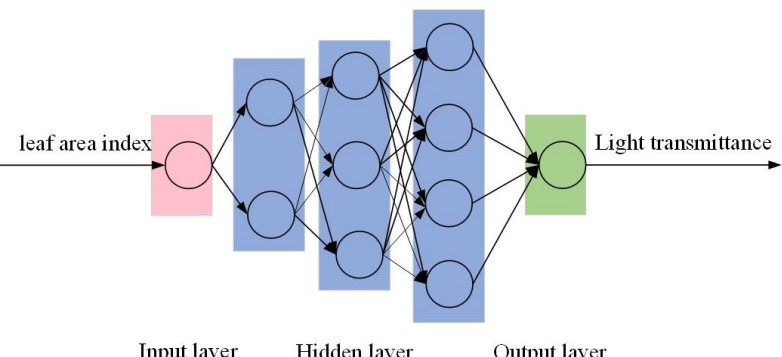

**Figure 7.** Schematic diagram of DNN.

The average sample size for training is 140, with a ratio of 0.8:0.1:0.1 for the training set, validation set, and test set, respectively.

The relationship between the Leaf Area Index and photosynthetically active radiation light transmittance is negatively exponential [40]:

$$PAR_{Ratio} = A \cdot e^{-K \times cLAI} \tag{14}$$

*A* is the correlation coefficient and *K* represents the extinction coefficient, which represents the light interception capacity.

## 3. Results

### 3.1. Point Cloud Preprocessing

By preprocessing the point cloud data collected from 35 fruit trees, we obtain the canopy point cloud information of the fruit trees, facilitating subsequent voxel processing and Leaf Area Index calculations. Using tree No. 21 in the orchard as an illustration, the effect of point cloud preprocessing is demonstrated in Figure 8.

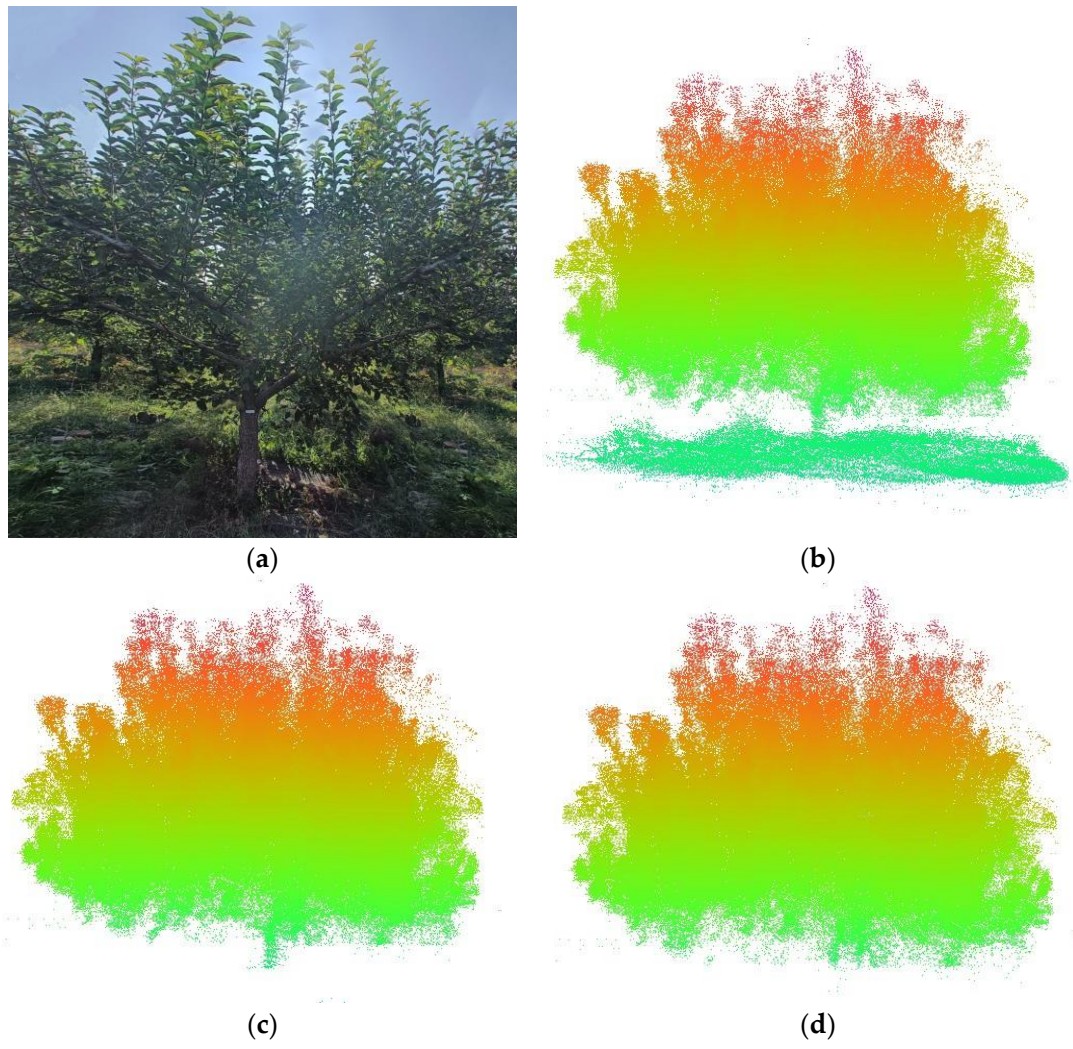

**Figure 8.** Renders from point cloud preprocessing. (**a**) Original image of tree No. 21. (**b**) Single wood segmentation. (**c**) Ground filtering. (**d**) Backbone removal.

In Figure 8a, the original image of tree No. 21 is displayed, chosen for its distinct main trunk, abundant branches and leaves, and well-defined crown outline, rendering it more representative than other trees. Figure 8b showcases the point cloud image of a single fruit tree, while Figure 8c exhibits the point cloud image of the fruit tree after ground filtering. Finally, Figure 8d presents the point cloud image with the branches removed.

### 3.2. Determination of Voxel Size

Applying the standard Euclidean distance formula, we calculate the average canopy point cloud spacing for each fruit tree and the mean of these average point cloud spacings, as depicted in Figure 9a. The red dashed line in the figure represents the mean value line of the average point cloud spacing (y = 0.015 m). Concurrently, a statistical analysis is conducted on the average point cloud spacing of each fruit tree; the distribution histogram is presented in Figure 9b. From Figure 9b, it is evident that among all the scanned fruit trees, 15 trees have an average point cloud spacing ranging from 0.015 to 0.0165 m. This indicates that the mean value of the average point cloud spacing effectively represents the distance of the canopy point clouds of the fruit trees. Therefore, for subsequent Leaf Area Index calculations, a voxel size of 0.015 m is selected.

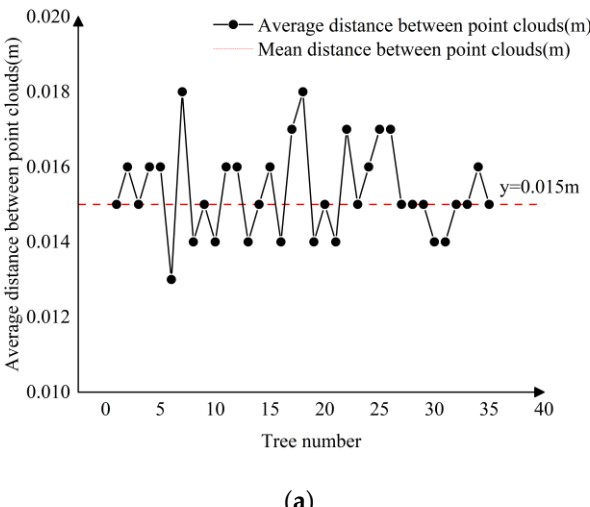 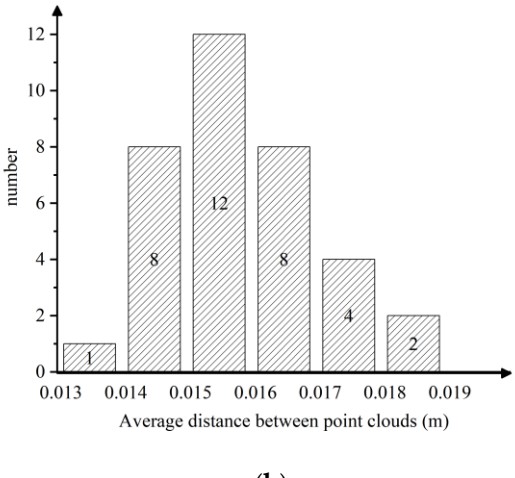

(**a**)  (**b**)

**Figure 9.** The average distance between point clouds of all tree data. (**a**) Point cloud average spacing value and *Y*-axis mean line. (**b**) Distribution histogram.

### 3.3. Calculation of Leaf Area Index

Thirty sets of leaf point clouds were randomly selected, with each set comprising 650 points. These points were used to fit the plane of the leaf using the Eigenvector method, allowing for estimation of the average orientation of the leaf. The angle between this average orientation and the zenith direction determined the leaf inclination angle. The distribution of these leaf inclination angles is illustrated in Figure 10. From Figure 10, it can be observed that the distribution of leaf inclination angles for apple trees ranges from 0 to 50 degrees, with the highest concentration falling within the 15–20-degree interval. The average leaf inclination angle is measured at 25.14 degrees.

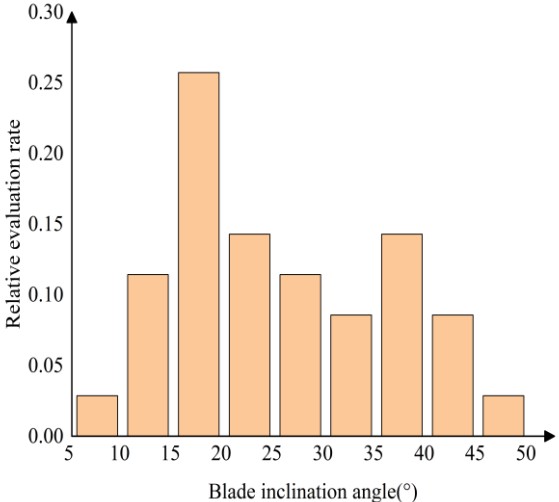

**Figure 10.** Frequency distribution diagram of blade inclination angle.

Formula (4) is employed to derive the zenith angle through coordinate transformation. The distribution of zenith angles spans from 3.6° to 33.8°, covering a range of approximately 30 degrees. The average zenith angle is calculated to be 19.24°. Since the LiDAR used is handheld, it operates relatively close to the target tree. The correction coefficient is determined using a specific formula. For zenith angles $\theta$ less than or equal to 90°, the correction factor remains constant at 0.91. However, when the zenith angle $\theta$ exceeds 90°, the correction factor varies with x.

The model's voxel size is set to 0.015 m, and a fixed horizontal layer thickness (ΔH = 0.5 m) is utilized to calculate the canopy contact frequency. Figure 11 illustrates that the contact frequency reaches its maximum value at a tree height of 1.5 m. This peak suggests that the leaves are most densely packed at this height, resulting in the highest contact frequency.

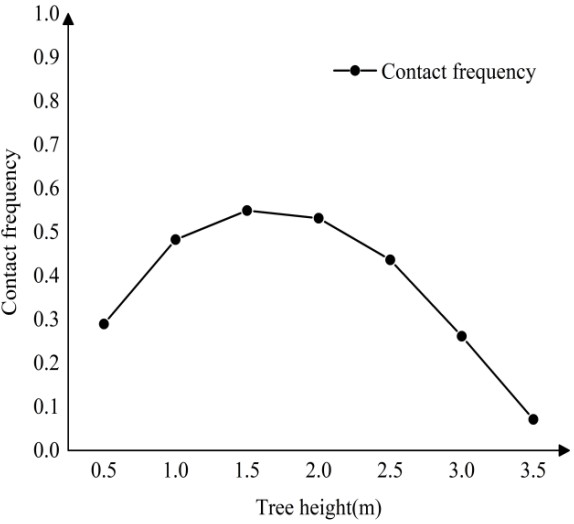

**Figure 11.** Contact frequency of canopy at different heights.

A fruit tree VCP (Voxel Crown Profile) model with a voxel size of 0.015 m was established. The majority of measured Leaf Area Index (LAI) values range from 0.9 to 1.2 $m^2/m^3$, with a maximum measured value of 1.89 $m^2/m^3$ and a minimum value of 0.75 $m^2/m^3$. Most reverse estimated LAI values fall within the range of 0.95 to 1.3 $m^2/m^3$. The maximum inverse estimate is 2.2 $m^2/m^3$, while the minimum value is 0.75 $m^2/m^3$. Figure 12 presents a comparison between the measured and inverse estimated values of Leaf Area Index. Explanation of the figure: the pink box on the left displays the distribution of measured LAI values for 35 apple tree canopies, while the corresponding scatter plot on the right shows the specific values. The light blue box on the right displays the distribution of reverse LAI values, with the corresponding scatter plot showing the specific values of the same 35 apple tree crowns.

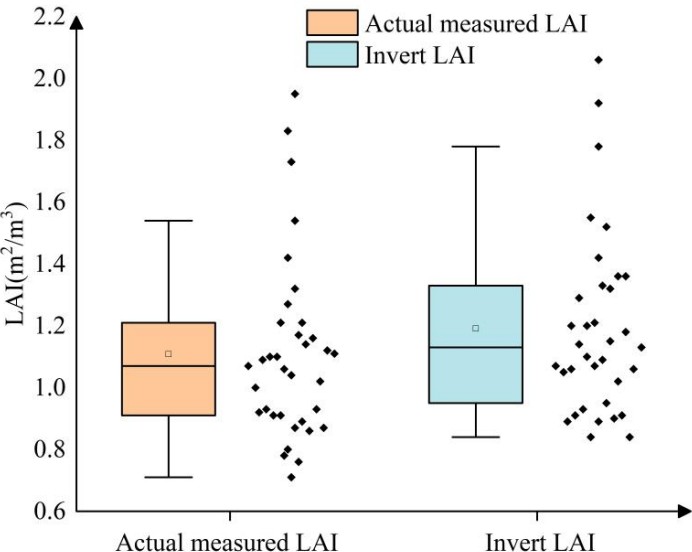

**Figure 12.** Comparison between measured LAI and inverted LAI.

Figure 11 indicates that the measured LAI values are generally lower than the inverted LAI values. To accurately assess the discrepancy, we utilize the root-mean-square-error (RSME) statistical index, which is a common measure of the error magnitude between predicted and actual values. The smaller the RSME, the closer the predicted value is to the actual value, indicating a more accurate prediction model. After precise calculation, the measured LAI and inverted LAI both have an RSME of 0.14.

### 3.4. Calculation of Light Transmittance

The assessment of fruit tree canopy light transmittance is primarily conducted using a canopy analyzer. Figure 13 displays the light transmittance results of 35 selected fruit trees. These trees are arranged based on their Leaf Area Index (LAI), with LAI depicted in blue-green and light transmittance in pink. It is noteworthy that there exists a negative correlation between the Leaf Area Index and canopy light transmittance. Specifically, among the apple trees in the orchard, tree No. 21 exhibits the highest Leaf Area Index concurrent with the lowest light transmittance. Tree No. 31, on the other hand, has the lowest Leaf Area Index and the highest light transmittance.

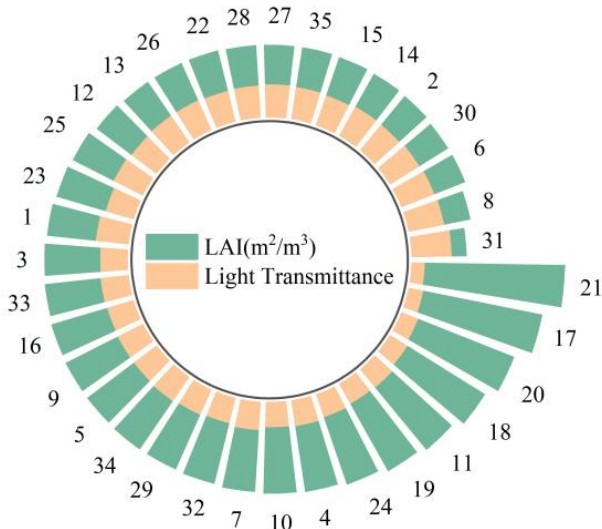

**Figure 13.** LAI–light transmittance chart.

### 3.5. Fitting Based on DNN Leaf Area Index and Light Transmittance

As shown in Figure 14a, it is an intuitive representation of the model convergence process.After 18 runs, the best results were obtained in round 12, yielding a mean square error (MSE) of 0.005. The model's overall mean absolute error (MAE) is 0.014, and the root mean squared error (RMSE) is 0.024. As shown in Figure 14b, a visual representation of the model results, with the training set represented by a blue line, the validation set by a green line, and the test set by a red line. The intersection line, denoted by a dashed line, marks the optimal training point, achieving an overall correlation coefficient ($R^2$) of 0.94.

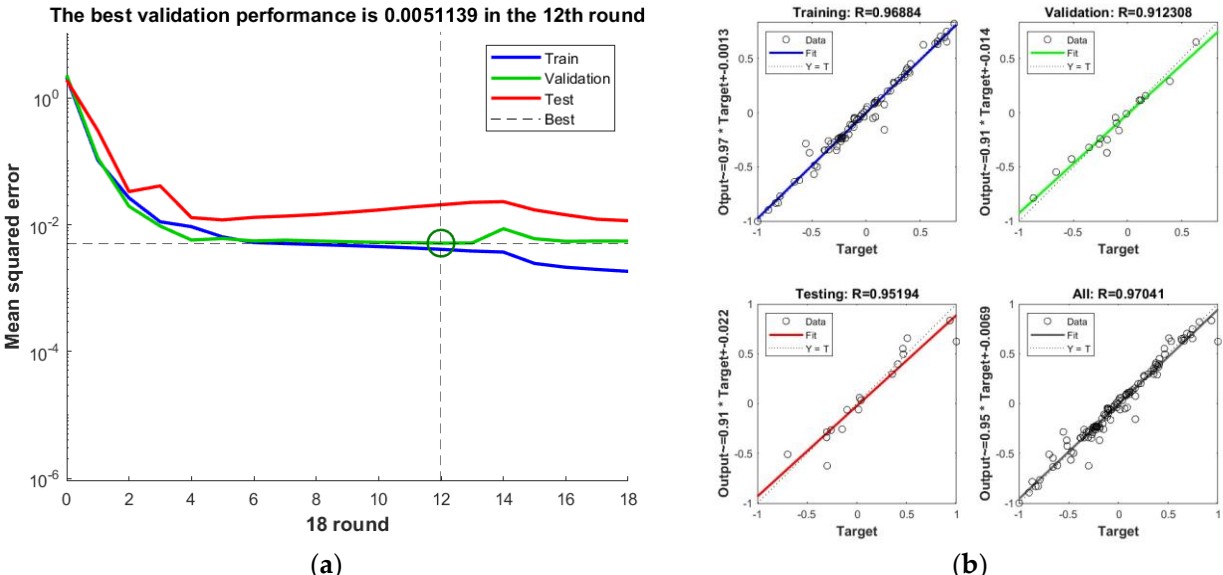

(**a**)                                                    (**b**)

**Figure 14.** Result graph based on DNN model. (**a**) Model convergence process. (**b**) Model correlation coefficient graph.

## 4. Discussion

(1)    Factors Affecting Light Transmission Analysis

As depicted in Figure 13, the Leaf Area Index (LAI) exerts a considerable influence on canopy light transmittance. Exploring the mechanism through which LAI affects canopy light transmittance is imperative for enhancing canopy nutrient accumulation and enhancing fruit quality. To delve deeper into the relationship between LAI and light transmittance, a scatter plot of LAI against the corresponding light transmittance for all fruit trees is generated to observe the trend. This scatter plot is illustrated in Figure 15.

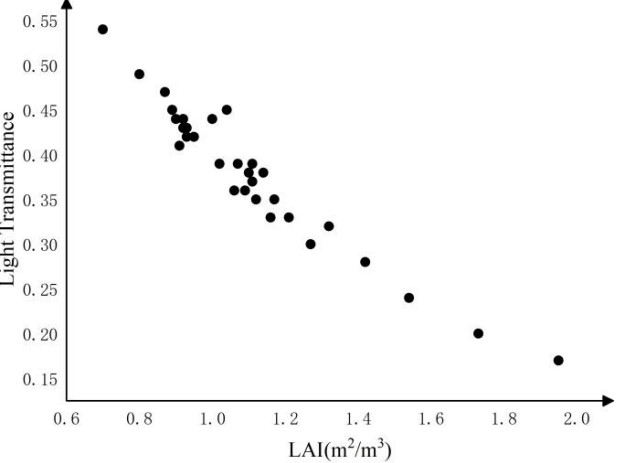

**Figure 15.** LAI–light transmittance scatter plot.

The correlation between the Leaf Area Index (LAI) in the apple tree canopy and light transmittance is highly significant and exponential in nature, as indicated by the following equation:

$$LT = 2.94\mathrm{e}^{-2.78LAI} \tag{15}$$

The trend in light transmittance within the apple tree canopy is observed to decrease as the Leaf Area Index increases. The larger the Leaf Area Index, the lower the light transmittance, and the higher the light utilization efficiency; the smaller the Leaf Area Index, the higher the light transmittance, and the lower the light utilization efficiency. Typically,

the Leaf Area Index of apple trees ranges from approximately 1.1 to 1.7 [41], ensuring an even distribution of canopy leaves and maintaining consistent levels of light transmittance.

The incident angle of light, extinction coefficient, and leaf distribution position also impact light transmittance, ultimately influencing yield. The Leaf Area Index serves as a general indicator of photosynthetic yield, making leaf distribution a crucial factor in determining light transmission rates. Trees exhibit varying angles of growth for their shapes, branches, and leaves. Leaves growing at more oblique angles have smaller leaf inclination angles, resulting in a higher Leaf Area Index and light interception. Conversely, when leaves grow closer to a flat orientation, the leaf inclination angle increases, leading to a lower Leaf Area Index. An optimized leaf structure enhances canopy ventilation and light transmission, thereby increasing the photosynthetic area. Further research is needed to delve into the issue of leaf growth positioning.

(2)     Comparison of Methods for Acquiring Light Transmittance

In this paper, the Leaf Area Index (LAI) was inverted using a point cloud, and canopy light transmittance was collected using a canopy analyzer. A model was then established as a function of LAI and canopy light transmittance. The correlation coefficient ($R^2$) of the model was 0.94, indicating a strong correlation between canopy light transmittance and LAI.

As depicted in Figure 16, the comparison between two methods of collecting light transmittance time is presented. The purple bar on the left represents the time required for light transmittance collection using the point cloud approach, while the red bar on the right represents the time taken for light transmittance collection using the canopy analyzer approach. The upper bar indicates the time needed for data analysis, whereas the lower bar denotes the time needed for data collection. To compare the duration of data collection, the average time required for light transmittance collection using the point cloud method was 3 min, whereas for light transmittance collection using the canopy analyzer, which involves collecting data from a single tree, it took 15 min. Regarding the duration of data analysis, it took 12 min for data preprocessing based on the point cloud method of light transmittance collection, and 20 min for data processing based on the canopy analyzer method of light transmittance collection. Experimental work involving canopy light transmittance measurement demands a significant amount of sample data and numerous readings, rendering it both time-consuming and labor-intensive. The point cloud data collection method based on LiDAR involves scanning a large area, thus proving more efficient when dealing with a larger number of fruit trees. Given that most apple orchards in China are planted in medium- to large-sized orchards, the efficiency of point cloud-based acquisition surpasses that of using canopy analyzers.

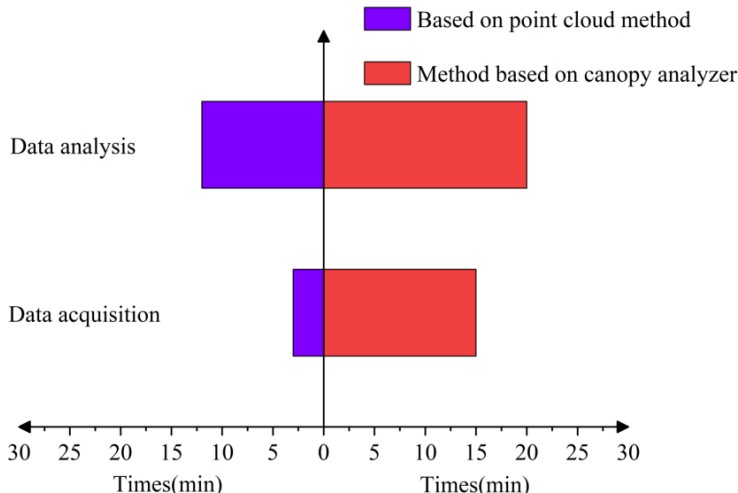

**Figure 16.** Comparison of the duration of various acquisition methods.

Canopy analyzers can pose challenges due to lighting issues and anthropogenic influences, leading to variability in subjective selection and reduced data accuracy. To address this, we conducted experimental data collection under controlled lighting conditions for specific time periods. This eliminated subjective factors such as the light angle, intensity, and anthropogenic influences in modeling light transmission and Leaf Area Index functions. This paper utilizes the Leaf Area Index to estimate light transmittance, enhancing objectivity by avoiding subjective errors in the measurement process.

In conclusion, the point cloud method offers a more efficient and objective way to measure light transmission compared to the two acquisition methods.

## 5. Conclusions

In this paper, a three-dimensional model of apple trees was constructed through canopy contour analysis (VCP). The canopy voxel size was determined based on the standard Euclidean distance, and the Leaf Area Index (LAI) was derived. Subsequently, an estimation model for LAI and canopy light transmittance was established. This model provides a theoretical basis for pruning fruit trees in orchards. Under certain light conditions, altering the canopy structure can modify the canopy Leaf Area Index, thereby increasing the light intensity received by the canopy and enhancing nutrient accumulation. However, it should be noted that this study focuses on flat fruit tree canopy research. If there are changes in tree species or canopy structure, the theoretical model may need to be adjusted accordingly.

**Author Contributions:** Conceptualization, L.Z. and F.K.; methodology, L.Z.; software, L.Z.; validation, S.T., C.C. and L.Z.; formal analysis, L.Z.; investigation, Y.W.; resources, F.K.; data curation, Y.W.; writing—original draft preparation, L.Z. and C.C.; writing—review and editing, F.K. and Y.W.; visualization, L.Z.; supervision, Y.W.; project administration, Y.W. and F.K.; funding acquisition, F.K. All authors have read and agreed to the published version of the manuscript.

**Funding:** This research was funded by the Ningxia Hui Autonomous Region key research and development plan project, grant number 2022BBF01002-02.

**Data Availability Statement:** All data generated or presented in this study are available upon request from the corresponding author. Furthermore, the models and code used during the study cannot be shared at this time as the data also form part of an ongoing study.

**Conflicts of Interest:** The authors declare no conflicts of interest.

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
