# Peer review of "Influence of Leaf Area Index Inversion and the Light Transmittance Mechanism in the Apple Tree Canopy"

_forests, doi:10.3390/f15050823_

Round 1
Reviewer 1 Report
Comments and Suggestions for Authors
The paper provides a detailed analysis of the methods used to measure canopy transmittance, including methods based on measuring photosynthetic radiation with instruments, ray tracing, and LiDAR scanning combined with the Beer-Lambert law.
Figure 14, change the Chinese to English.
Following two parts that you need further discussion.
The method using LAI-2200 under poor lighting conditions introduces issues like light obstruction and requires the use of a view cap to eliminate adverse factors, leading to increased operational complexity and subjective selection variability. This subjectivity can impact the reliability and consistency of the data collected, highlighting the need for more objective and standardized measurement techniques.
Experimental work on measuring canopy transmittance requires a large amount of sampling data and numerous readings, making it both time-consuming and labor-intensive. Developing automated or semi-automated data collection and analysis processes could help streamline the research workflow and reduce the burden of manual data collection and processing.
Comments on the Quality of English Language
Figure 14
Author Response
forests-2934535
Dear Reviewer:
Thank you for your Review and Specific Comments concerning our manuscript “Research on the Inversion of Leaf Area Index and the Mechanism of Light Transmittance Influence in Apple Tree Canopy”.
As you pointed out, the first draft of this paper contains a number of poorly articulated points. I have carefully addressed the issues you raised and have added to and improved the paper accordingly. After making changes based on your suggestions, the overall quality of the article has improved significantly, which will help both the readability of the article and my future writing efforts.If there are still deficiencies, we hope to get another opportunity to modify it.
Thank you very much.
Yours sincerely,
Linghui Zhou
Beijing Forestry University
Point 1:Figure 14, change the Chinese to English.
Reply 1: Thank you very much for your guidance and advice.The error in Figure 14b, where Chinese characters appeared, has been corrected. Please refer to lines 466-471 of the article for further details.
Point 2:Following two parts that you need further discussion.
The method using LAI-2200 under poor lighting conditions introduces issues like light obstruction and requires the use of a view cap to eliminate adverse factors, leading to increased operational complexity and subjective selection variability. This subjectivity can impact the reliability and consistency of the data collected, highlighting the need for more objective and standardized measurement techniques.
Reply 2: Thank you very much for your guidance and advice.To estimate light transmission using a canopy analyzer, external factors must be taken into account. Therefore, we conducted the experiment during the same time period, under windless and sunny weather conditions. In this paper, we used the leaf area index to estimate transmittance, avoiding any subjective errors that may occur during the measurement process. However, I neglected to explain the objectivity of this study in the discussion section. Therefore, I have revised the section and added a discussion on the objectivity of data collection. Please refer to lines 541-550 of the article for more details.
Point3:Experimental work on measuring canopy transmittance requires a large amount of sampling data and numerous readings, making it both time-consuming and labor-intensive. Developing automated or semi-automated data collection and analysis processes could help streamline the research workflow and reduce the burden of manual data collection and processing.
Reply 3: Thank you very much for your guidance and advice. Canopy analyzers are commonly used to measure transmittance, but manual work can be time-consuming and laborious. Our point cloud-based approach improves efficiency by collecting the point cloud as a whole piece, reducing data collection time. However, the discussion section does not address this efficiency improvement. Based on your feedback, we have made revisions and included two types of analysis regarding the time it takes to acquire light transmittance in the discussion section. Please refer to lines 522-536 of the article for more information.

Reviewer 2 Report
Comments and Suggestions for Authors
1. What is the main question addressed by the research?
The study “Research on the Inversion of Leaf Area Index and the Mechanism of Light Transmittance Influence in Apple Tree Canopy” by LingHui Zhou et. al. is aimed to estimate LAI of an apple tree by means of the author's Voxel-based Canopy Profile model method to invert the canopy Leaf Area Index.
2. What parts do you consider original or relevant for the field? What specific gap in the field does the paper address?
The model was developed that uses LiDAR point cloud data in the Voxel-based Canopy Profile model method to invert the canopy Leaf Area Index.
3. What does it add to the subject area compared with other published material?
The study contains the description of the novel combined method of LAI estimation and also some values of LAI for the real apple tree.
4. What specific improvements should the authors consider regarding the methodology? What further controls should be considered?
The authors should explain in more detail how the “magic” works inside the “Hidden layers” (Figure 7) and also the Eigenvector Method, mentioned in Line 366.
5. Please describe how the conclusions are or are not consistent with the evidence and arguments presented. Please also indicate if all main questions posed were addressed and by which specific experiments.
The obtained LAI estimates are fully consistent with reality. To collect the point cloud data the scanning of the trees was performed. A fruit tree model was established and the inverse-estimated values of LAI was obtained by use of neural network.
6. Are the references appropriate?
In the Introduction there are mentioned “methods based on measuring photosynthetic radiation with instruments, methods based on ray tracing, and methods that combine Li-DAR scanning with the Beer-Lambert law”. Each mentioned method needs a reference.
7. Please include any additional comments on the tables and figures and quality of the data.
There is not indicated what does red dotted line in Fig. 9a mean.
In Lines 368-369 must be reference to Figure 10 (not 9)
In Figure 12: what do the dots mean? What are rectangles?
In Lines 406, 408 there is a reference to non-existent Figure 16.
What is the difference between the dotted lines “Best” and “Goal” in Figure 14 a?
What is the difference between the graphs in Figure 14 b? The inscriptions are in Chinese.
To my mind the article can be accepted for publication in "Forests" after making the indicated corrections.
Author Response
forests-2934535
Dear Reviewer:
Thank you for your Review and Specific Comments concerning our manuscript “Research on the Inversion of Leaf Area Index and the Mechanism of Light Transmittance Influence in Apple Tree Canopy”.
As you pointed out, the first draft of this paper contains a number of poorly articulated points. I have carefully addressed the issues you raised and have added to and improved the paper accordingly. After making changes based on your suggestions, the overall quality of the article has improved significantly, which will help both the readability of the article and my future writing efforts.If there are still deficiencies, we hope to get another opportunity to modify it.
Thank you very much.
Yours sincerely,
Linghui Zhou
Beijing Forestry University
Point 1:What is the main question addressed by the research?
The study “Research on the Inversion of Leaf Area Index and the Mechanism of Light Transmittance Influence in Apple Tree Canopy” by LingHui Zhou et. al. is aimed to estimate LAI of an apple tree by means of the author's Voxel-based Canopy Profile model method to invert the canopy Leaf Area Index.
Reply 1:Thank you very much for your guidance and advice. This paper presents a method for estimating light transmission rate by leaf area index (LAI) using canopy point cloud data and a canopy analyzer. The LAI values were inverted and the canopy transmittance was collected to establish a function model of LAI and transmittance. The comparison between the theoretical model derivation and actual equipment observation highlights the advantages of this method.
Theoretical modeling can be used to estimate the canopy light transmission rate, which can reduce the impact of supervisor's operation errors during equipment measurement. This approach leads to more objective and scientific experimental results.
Additionally, using theoretical modeling to estimate canopy transmittance can significantly improve the efficiency of data collection.
The introduction has already mentioned the relevant information from the two parts. To enhance the logical flow and improve reader comprehension, the discussion section of the article provides further explanation and illustration. Refer to lines 522-550 for details.
Point 2:What parts do you consider original or relevant for the field? What specific gap in the field does the paper address?
The model was developed that uses LiDAR point cloud data in the Voxel-based Canopy Profile model method to invert the canopy Leaf Area Index.
Reply 2:Thank you very much for your guidance and advice.The objective of this paper is to estimate canopy light transmittance based on canopy leaf area index. In contrast to current research trends, this paper presents a theoretical model of leaf area index and light transmittance, clarifying the functional relationship between the two. In addition, the article emphasizes the scientific and efficient data acquisition advantages of the theoretical model by comparing it with the actual equipment observation. This introduction is detailed in lines 522-550 of the article.
Point 3:What does it add to the subject area compared with other published material?
The study contains the description of the novel combined method of LAI estimation and also some values of LAI for the real apple tree.
Reply 3:Thank you very much for your guidance and advice.Currently, research on the mechanisms that affect canopy light transmittance is dominated by trends. The focus is on highlighting the influence of leaf area index on canopy light transmittance, which is consistent with the presentation of this paper. Based on this study, a function model of leaf area index and light transmittance is constructed to make the relationship between the two more tangible.
Point 4:What specific improvements should the authors consider regarding the methodology? What further controls should be considered?
The authors should explain in more detail how the “magic” works inside the “Hidden layers” (Figure 7) and also the Eigenvector Method, mentioned in Line 366.
Reply 4:Thank you very much for your guidance and advice.This paper aims to construct a model for the LAI-transmittance function using recognized and effective methods to avoid any influence of the operation on the data. The voxel dimensions acquisition and light transmittance measurement are based on the actual situation of the orchard and the environmental conditions. The article employs the voxel method to reconstruct the canopy. The size of the voxel used during the reconstruction process affects the final reconstruction accuracy. The size of the voxel is determined by the reconstructed canopy species, and it needs to be re-determined when the type of plant is changed. The article acknowledges this limitation and provides further details in lines 355-357.
The article describes the process of canopy reconstruction and biomass information acquisition, while simplifying the model construction process due to space constraints. Additional explanations have been provided in lines 361~379 of the article to improve the reader's understanding and readability, as per the editor's suggestions.
The article explains the eigenvalue method for fitting the blade plane in a simplified manner. As per your comments, I have added further explanation of the calculation process for this part. Please refer to lines 274-287 of the article for more details.
Point 5: Please describe how the conclusions are or are not consistent with the evidence and arguments presented. Please also indicate if all main questions posed were addressed and by which specific experiments.
The obtained LAI estimates are fully consistent with reality. To collect the point cloud data the scanning of the trees was performed. A fruit tree model was established and the inverse-estimated values of LAI was obtained by use of neural network.
Reply 5:Thank you very much for your guidance and advice.This paper constructs a LAI-transmittance function model to establish the relationship between canopy characteristics and transmittance, simplifying the acquisition of transmittance and providing theoretical basis for subsequent pruning. The construction of this model addresses the drawbacks mentioned in the introduction, namely the cumbersome and labor-intensive process of measuring transmittance using instruments. Currently, scholars' analyses of the trend of LAI's influence on transmittance are consistent, which not only validates the accuracy of the trend but also elaborates on the relationship between them.
To establish the LAI-transmittance relationship, the article primarily undertakes two operations: (1) Three-dimensional reconstruction of the canopy using mainstream methods to obtain canopy characteristic values, i.e., LAI; (2) Utilizing canopy analyzers as the mainstream method to collect and acquire transmittance. These operations ensure the accuracy of the LAI and transmittance data. The article constructs a function model for Leaf Area Index (LAI) and transmittance. In the discussion section, a comparative analysis is made between the efficiency and scientificity of model calculations and equipment measurements, addressing the issue raised in the introduction regarding the subjective influence on equipment due to operation.
Point 6:Are the references appropriate?
In the Introduction there are mentioned “methods based on measuring photosynthetic radiation with instruments, methods based on ray tracing, and methods that combine Li-DAR scanning with the Beer-Lambert law”. Each mentioned method needs a reference.
Reply 6:Thank you very much for your guidance and advice.
After extensively reviewing the literature, the introduction has been drafted to summarize the findings. The content mentioned above has been supported with relevant citations as suggested by the editor. Please refer to lines 61-66, 81-83, and 93-96 of the article for detailed references.
Point 7:Please include any additional comments on the tables and figures and quality of the data.
There is not indicated what does red dotted line in Fig. 9a mean.
In Lines 368-369 must be reference to Figure 10 (not 9)
In Figure 12: what do the dots mean? What are rectangles?
In Lines 406, 408 there is a reference to non-existent Figure 16.
What is the difference between the dotted lines “Best” and “Goal” in Figure 14 a?
What is the difference between the graphs in Figure 14 b? The inscriptions are in Chinese.
To my mind the article can be accepted for publication in "Forests" after making the indicated corrections.
Reply 7:Thank you very much for your guidance and advice.
During the writing process, insufficient attention was given to the detailed handling of the images, and the identified issues indeed required modification. Following your suggestions, I have made the following corrections:
(1)In Figure 9(a), the red dashed line represents the mean line of the average point cloud spacing of 35 fruit trees. To enhance readability, I have added corresponding legends to the figure. Please refer to lines 415-416 of the article for details.
(2)In the process of inserting figures into the article, there were errors in the numbering. I have rectified this according to the issues raised by you. Please see lines 417-418 of the article.
(3)Figure 12 depicts the comparison between measured leaf area index and inverted leaf area index. The article now includes relevant explanations for this figure. Please refer to lines 454-458 of the article.
(4)As mentioned in point (2), there were numbering errors in the process of inserting figures into the article, which have now been rectified.
(5)Following your suggestions, I have changed the linear style in Figure 14(a) and provided explanations. Please see lines 482-487 of the article for details.
(6)There were Chinese characters in Figure 14b due to image discrepancies. This issue has been rectified. Please refer to lines 486-487 of the article for details.

Reviewer 3 Report
Comments and Suggestions for Authors
The paper "Research on the Inversion of Leaf Area Index and the Mechanism of Light Transmittance Influence in Apple Tree Canopy" is an interesting research topic that can be improved by modifying it according to the following remarks:
1-As I understand from figure 5 that the authors collects the Lidar data for each tree separately. Is this an efficient and reliable way to manage large fields of fruit trees or forests?
Is there any other way to collect Lidar data for large forests or orchards? Please explain.
2-There are many issues related to the use of the equations 2 to 8.
A- The origin of these equations (references)
B- Same symbols are used for different purposes such as theta
3-The authors should give more details about the use of the artificial neural network to predict light transmittance. Please provide the name of the network, type of the network,...etc.
4- How the authors obtained equation 13?
5- Does the input to the network represent the leaf area of multiple trees or one tree?
6-Subsection 3.3 should be part of the data collection in data and methods section.
7- Calculate the RMSE for the measured and inverted LAI.
8- It is better if the graph in Figure 12 shows measured vs estimated LAI.
Author Response
forests-2934535
Dear Reviewer:
Thank you for your Review and Specific Comments concerning our manuscript “Research on the Inversion of Leaf Area Index and the Mechanism of Light Transmittance Influence in Apple Tree Canopy”.
As you pointed out, the first draft of this paper contains a number of poorly articulated points. I have carefully addressed the issues you raised and have added to and improved the paper accordingly. After making changes based on your suggestions, the overall quality of the article has improved significantly, which will help both the readability of the article and my future writing efforts.If there are still deficiencies, we hope to get another opportunity to modify it.
Thank you very much.
Yours sincerely,
Linghui Zhou
Beijing Forestry University
Point 1:As I understand from figure 5 that the authors collects the Lidar data for each tree separately. Is this an efficient and reliable way to manage large fields of fruit trees or forests?
Is there any other way to collect Lidar data for large forests or orchards? Please explain.
Reply 1:Thank you very much for your guidance and advice.
Figure 5 illustrates a top-down schematic diagram of canopy light transmittance acquisition, with measurements of individual canopy light transmittance taken according to the positions indicated in the diagram. Please refer to lines 218-220 of the article for details.
There are two main methods for acquiring point cloud data: those based on laser radar and those based on imagery. Using image-based methods has its limitations, including constraints on perspective, shadows, and reflections, and high requirements for external environmental conditions. Depth information needs to be extracted as a preprocessing step. On the other hand, point cloud data acquisition using laser radar is not affected by lighting conditions and offers direct and accurate processing. In this study, point cloud data are utilized for the inversion of LAI values to construct the LAI-transmittance function model. Hence, there is a high demand for canopy biomass information. This aspect is mentioned in the introduction section of the article; please see lines 48-54 for further details.
Point 2:There are many issues related to the use of the equations 2 to 8.
A-The origin of these equations (references)
B- Same symbols are used for different purposes such as theta
Reply 2: Thank you very much for your guidance and advice.
- According to the references consulted, additional theoretical explanations have been added to the theoretical parts of Equations 2 to 8. Please refer to line 292 of the article for details.
- In this chapter, θoriginally referred to the solar zenith angle, while θLrepresented the leaf inclination angle. Following your suggestion, I have made modifications. To avoid confusion among readers regarding the meaning of symbols, θL has been changed to the symbol ψ. Please see lines 294-296 of the article for details.
Point 3:The authors should give more details about the use of the artificial neural network to predict light transmittance. Please provide the name of the network, type of the network,...etc.
Reply 3: Thank you very much for your guidance and advice.
The article primarily focuses on describing the process of canopy reconstruction and biomass information acquisition. Due to space constraints, the model construction process was simplified. To facilitate reader comprehension and enhance the readability of the article, additional explanations have been provided in accordance with the editor's suggestion. Please refer to lines 361-379 of the article for details.
Point 4:How the authors obtained equation 13?
Reply 4:Thank you very much for your guidance and advice.
Equation 13 is used to determine the number of hidden layers, where 'a' represents the adjustable variable, 'bi' represents the number of input variables, and 'bp' represents the number of output variables. Corresponding explanations can be found in lines 383-384 of the article. This equation is introduced to address the determination of the number of hidden layers in the DNN model, and this aspect is derived from references [35] and [36], which are appropriately cited in the text.
Point 5:Does the input to the network represent the leaf area of multiple trees or one tree?
Reply 5: Thank you very much for your guidance and advice.
In order to improve the accuracy of the model, information from 35 fruit trees was collected. To facilitate reader understanding, corresponding explanations have been added in the text. Please refer to line 366 of the article for details.
Point 6:Subsection 3.3 should be part of the data collection in data and methods section.
Reply 6:Thank you very much for your guidance and advice.
Section 3.3 of the article presents the inversion results of the canopy leaf area index. The introduction of the inversion method and process is already explained in the Materials and Methods section of the article. Please refer to lines 262-302 for details. The presentation of the results in this section strictly follows the narrative outlined in the Materials and Methods, ensuring a clear correspondence between the method and the results.
Point 7:Calculate the RMSE for the measured and inverted LAI.
Reply 7:Thank you very much for your guidance and advice.
Figure 12 shows the comparison between the measured leaf area index and the inverted leaf area index. The RMSE values between the two can be reflected in the figure. In order to make the error more intuitive, while retaining Figure 11, the overall error between the calculated and inverted values of the leaf area index of 35 fruit trees was calculated, and this part of the content was described in the article. Please refer to lines 459-464 for details.
Point 8:It is better if the graph in Figure 12 shows measured vs estimated LAI.
Reply 8:Thank you very much for your guidance and advice.
Figure 12 shows the comparison between the measured leaf area index and the inverted leaf area index. I have drawn histograms and box plots respectively. Due to the large amount of data, although histograms can intuitively express measurement data, the expression effect is quite confusing, which affects the readability of the article. Of course, it is necessary to describe the error between the measured and inverted values of leaf area index. Therefore, I have provided additional explanations in this section of the article, as detailed in lines 454 to 464.

Round 2
Reviewer 3 Report
Comments and Suggestions for Authors
The authors have done good job in improving their paper according to the reviewer's remarks. However, there one more question not answered correctly. The author can add the answer in the discussion or the conclusion.
1- As I understand from figure 5 that the authors collects the Lidar data for each tree separately. Is this an efficient and reliable way to manage large fields of fruit trees or forests?
This question can be formulated differently: How do you manage large area of forests using this technique?
Comments on the Quality of English Language
There are few grammatical and vocabulary errors. A revision of the English language structure would help increase the readability of the paper.
Author Response
forests-2934535
Dear Reviewer:
Thank you for your Review and Specific Comments concerning our manuscript “Research on the Inversion of Leaf Area Index and the Mechanism of Light Transmittance Influence in Apple Tree Canopy”.
As you pointed out, the first draft of this paper contains a number of poorly articulated points. I have carefully addressed the issues you raised and have added to and improved the paper accordingly. After making changes based on your suggestions, the overall quality of the article has improved significantly, which will help both the readability of the article and my future writing efforts.If there are still deficiencies, we hope to get another opportunity to modify it.
Thank you very much.
Yours sincerely,
Linghui Zhou
Beijing Forestry University
Point1:1- As I understand from figure 5 that the authors collects the Lidar data for each tree separately. Is this an efficient and reliable way to manage large fields of fruit trees or forests?
This question can be formulated differently: How do you manage large area of forests using this technique?
Reply1:This article uses a laser radar to collect point cloud data of fruit trees, as shown in Figure 4. Based on the collected point cloud data, the canopy leaf area index is calculated. Figure 5 shows the collection of canopy light transmittance of fruit trees, which is obtained using a canopy analyzer to serve as the true value of canopy light transmittance. A function model between the canopy leaf area index and light transmittance is constructed based on the experimental steps in Figures 4 and 5.
The laser radar used in this study is a handheld scanning device. Due to limitations in its usage, the scanning area is typically controlled within a range of 2000 m². When the planting area of fruit trees is too large, an airborne laser radar needs to be used for scanning. The point cloud acquisition method is shown in Figure 4. A total of 35 fruit trees were scanned in this study, with a relatively small planting area, typical of common planting scales in China. Therefore, a handheld laser radar was used.
The canopy analyzer model used in this study is LD-G20H, which is commonly used for observing the canopy light transmittance of fruit trees. The observation method is shown in Figure 5. To avoid interference from the main trunk of the fruit tree, the canopy light transmittance was observed from four directions, each capturing a 90° range of observations. The results were then stitched together to calculate the light transmittance of the entire canopy. This method is currently the most commonly used method for measuring canopy light transmittance.
By using laser radar to collect point cloud information of orchards and following the procedures outlined in the article for filtering, single tree segmentation, and voxel reconstruction, the biomass information of each fruit tree was obtained. Then, the transmittance of each fruit tree canopy was calculated based on the light transmittance model constructed in the article. This method allows for the estimation of canopy light transmittance of orchard fruit trees, providing guidance for subsequent pruning operations. Moreover, it simplifies the process of obtaining canopy light transmittance of fruit trees and reduces the subjective errors compared to manually collecting light transmittance with equipment.
In the conclusion section of the article, the guidance significance of this study for orchard management is mentioned. Please refer to lines 552-561 of the article for details.
